# Increasing Data Efficiency of Driving Agent By World Model

**Qianqian CAO**
Department of Information Engineering
The Chinese University of Hong Kong
Shatin, Hong Kong
1155145521@link.cuhk.edu.hk

**Yicheng LIU**
Department of Information Engineering
The Chinese University of Hong Kong
Shatin, Hong Kong
1155152886@link.cuhk.edu.hk

## Abstract

Reinforcement learning algorithms for real-world autonomous driving must be able to handle complex, unknown dynamical systems. This requirement is handled well by model-free algorithm such as PPO. However, model-free approach tend to be substantially less sample-efficient. In this work, we aim to retain the advantages of model-free method and increase the stability and data-efficiency of PPO. To this end we proposed a world model to model popular reinforcement learning environments through compressed spatio-temporal representations, which allow model-free method learning behaviors from imagined outcomes to increase sample-efficiency. The experimental results indicate that our approach mitigating the inefficiency of PPO, increasing the stability, and largely reducing the training time. code is available at `www.github.com/Mrmoore98/World-Model.git`. The video can be found here.

## 1 Introduction

Driving agent for real-world autonomous driving should achieved goals in complex environments even though they never encounter the exact same situation twice. To plan in unknown environments, the agent needs to learn the dynamics from experience and generalize to the handle the unknown dynamic. The Model-free methods have the advantage of handling arbitrary dynamical systems with minimal bias. With these merits model-free method has been deployed in several application and achieved promising results. However, some of the best model-free reinforcement learning algorithms require tens or hundreds of millions of time steps – the equivalent of several weeks of training in real time, even in the simple environment such as cCarracing-V0[1] it still needs nearly four hour to converge into a optimal solution with single GPU. How is it that humans can learn these games so much faster? Perhaps part of the puzzle is that humans possess an intuitive understanding of the physical processes that are represented in the game and we can foresee the consequence of each action and make an optimal decision. Besides, learning policy directly from pixel is inefficient and costly. For instance, PPO approach the cCarracing-V0 environment need to use a stack of CNN units to scan the whole image and process the info summarized by the CNN which are costly since only the few attributes(e.g., velocity and position of car) are meaningful for the action decision, other

---

[1] For simplicity, we will conduct analysis in the cCarracing-V0 environments. Although It is a very simple environment, the issue of model-free method we claimed is still there.

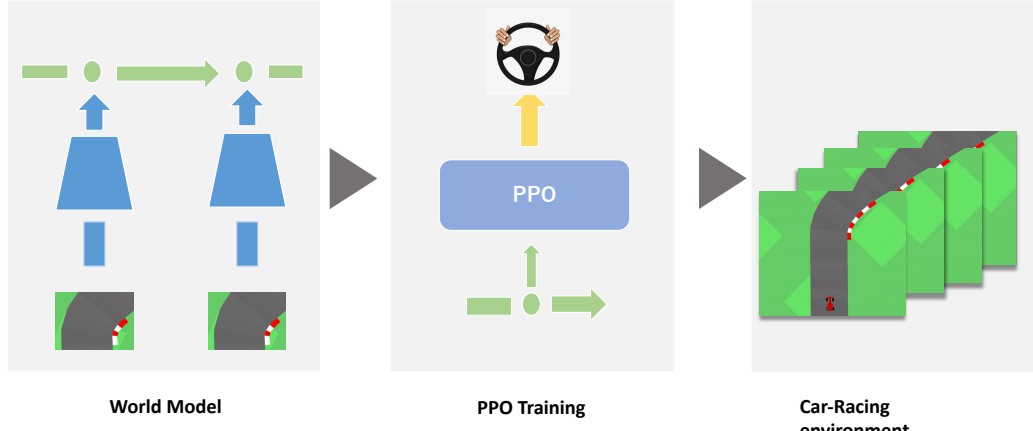

**World Model**  **PPO Training**  **Car-Racing environment**

Figure 1: Our model is an extension of the vanilla PPO. We modify the model of vanilla PPO by replace the observation input of control with a compact latent feature. This setting improve the model performance and increase the stability.

irrelevant information will contaminate the feature learned by policy gradient. To filter out these noise model will waste a large amount sample retrieved from interaction with environment. It is costly and data inefficient.

Planning using learned models offers several benefits over model-free reinforcement learning. First, model-based planning can be more data efficient because it leverages a richer training signal and does not require propagating rewards through Bellman backups. Moreover, planning carries the promise of increasing performance just by increasing the computational budget for searching for actions. Finally, learned dynamics can be independent of any specific task and thus have the potential to transfer well to other tasks in the environment.

Leveraging the advantage of model-based method we proposed a world model approach to alleviate the data inefficiency issue of PPO and accelerate the training time. A actor critic algorithm accounts for rewards beyond the imagination horizon while making efficient use of the neural network dynamics. For this, we predict latent state and future reward in the learned latent space by world model. The future discounted rewards is optimize Bellman consistency for latent rewards. the final policy is made conditioned on future rewards and latent state and optimized by PPO algorithm. Under this setting, policy can be learned with compact and informative feature which alleviate the optimization burden and increasing the efficiency of data utilization.

The key contributions of this paper are summarized as follows:

- **Increasing the data efficiency of model-free methods,** world models can interpolate past experience and provide predicted future feature for efficient policy optimization. our model can reach the maximum mean reward of vanilla PPO model only need 20% of original training time.

- **Reducing the total training time for the driving agent,** We using world model to distill a abstract, compressed representation from each observed input frame. Enabling us to design a low complexity parametric policy, which in our case is a two layer MLP. Therefore, our model having lesser update time and longer sample trajectories under same memory constrains.

- **Empirical results show our method is out performance to vanilla PPO,** We pair our method with vanilla PPO and evaluate it on the cCarracing-V0 environment with image inputs, illustrated in Figure 5. Two method using the same hyper parameters, our method exceeds PPO trained agents in terms of data-efficiency, computation time, and final performance.

## 2   Related Works

### 2.1   Control with latent dynamics

Prior works learn latent dynamics for visual control by derivative-free policy learning or online planning, augment model-free agents with multi-step predictions, or use analytic gradients of Q-values or multi-step rewards, often for low-dimensional tasks. In comparison, Dreamer uses analytic gradients to efficiently learn long-horizon behaviors for visual control purely by latent imagination.

Control with latent dynamics E2CWatter et al. [2015] and RCEBanijamali et al. [2018] embed images to predict forward in a compact space to solve simple tasks. World ModelsHa and Schmidhuber [2018] learn latent dynamics in a two-stage process to evolve linear controllers in imagination. PlaNetHafner et al. [2019a] learns them jointly and solves visual locomotion tasks by latent online planning. SOLARZhang et al. [2019] solves robotic tasks via guided policy search in latent space. I2A (Weber et al., 2017) hands imagined trajectories to a model-free policy, whileLee et al. [2020] and Gregor et al. [2019]learn belief representations to accelerate model-free agents.

### 2.2   Policy gradient Methods

Analytic value gradients DPG Silver et al. [2014], DDPG Lillicrap et al. [2015], and SACHaarnoja et al. [2018] leverage gradients of learned immediate action values to learn a policy by experience replay. SVG Heess et al. [2015] reduces the variance of model-free on-policy algorithms by analytic value gradients of one-step model predictions. Concurrent work by Byravan et al. [2020] uses latent imagination with deterministic models for navigation and manipulation tasks. ME-TRPO Kurutach et al. [2018] accelerates an otherwise model-free agent via gradients of predicted rewards for proprioceptive inputs. DistGBP Henaff et al. [2017] uses model gradients for online planning in simple tasks.

The structure of the model-based RL algorithm that we employ consists of alternating between learning a model, and then using this model to optimize a policy with model-free reinforcement learning. Variants of this basic algorithm have been proposed in a number of prior works, starting from Dyna Q Sutton [1991] to more recent methods that incorporate deep networks Kurutach et al. [2018], Heess et al. [2015], Feinberg et al. [2018], Kalweit and Boedecker [2017].

## 3   Method

To solve unknown environments efficiently, we need to model the environment dynamics from experience. our method does so by iteratively collecting data using random action and learning the latent dynamics of environment from the gathered data. we introduce notation for the environment and describe the general implementation of our model-based agent. In this section, we assume access to a learned dynamics model. Our design and training objective for this model are detailed in Section 4.

### 3.1   Problem Formulation

Problem setup Since individual image observations generally do not reveal the full state of the environment, we consider a partially observable Markov decision process (POMDP). We define a discrete time step $t$, hidden states $s_t$, image observations $o_t$, continuous action vectors $a_t$, and scalar rewards $r_t$, that follow the stochastic dynamics

$$\text{Representation function}: \ s_t \sim p(s_t|o_t) \tag{1}$$

$$\text{Transition function}: \ s_t \sim p(s_{\hat{t+1}}|s_t, a_t) \tag{2}$$

$$\text{Observation function}: \ o_t \sim p(o_t|s_t) \tag{3}$$

$$\text{Reward function}: \ r_t \sim p(r_t|s_t, a_t) \tag{4}$$

$$\text{Policy function}: \ a_t \sim p(a_t|a_{<=t}, o_{<=t}) \tag{5}$$

Without loss of generality, we assume initial state $s_0$ is fixed. The goal is to learn a policy $p(a_t|a_{<=t}, o_{<=t})$ that maximizes the expected sum of rewards $E_p[\sum_{t=1}^{T} r_t]$, where the expectation is over the distributions of the environment and the policy.

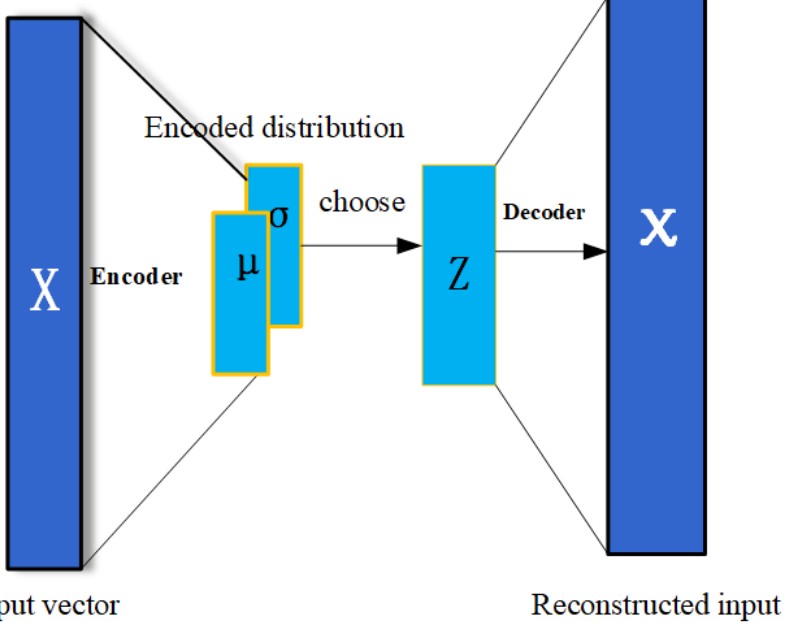

Figure 2: The diagram of VAE. The observations collected from the environment are summarized by a stack of CNN units and producing the final latent feature $z$. During these process, the size of observations $o_t$ are shrink from the $96 \times 96 \times 4$ into vector $s_t$ with only 256 elements.

## 3.2 Environment Encoder

Since Variational Auto-Encoder(VAE) is widely applied in image reconstruction due to its ability of maintaining the most essential features of inputs while discarding the irrelevant information. These property enable us to compress the observed image into a compact latent feature whose size is far smaller compare to the observed image as we show in Fig 2. Following Ha and Schmidhuber [2018] We design a CNN-based VAE as the representation function $s_t \sim p(s_t|o_t)$ and observation function $o_t \sim p(o_t|s_t)$. In particular, we define the encoder of the VAE as the representation function and decoder as the observation function, respectively. Based on these setting our model can generate a compact latent feature with sufficient environmental details, and the reconstruction stage is further guarantee the model the mutual information between the latent state $s_t$ and original observation $o_t$. These components are trained jointly to increase the variational lower bound Jordan et al. [1999] or more generally the variational information bottleneck Tishby et al. [2000], Alemi et al. [2016]

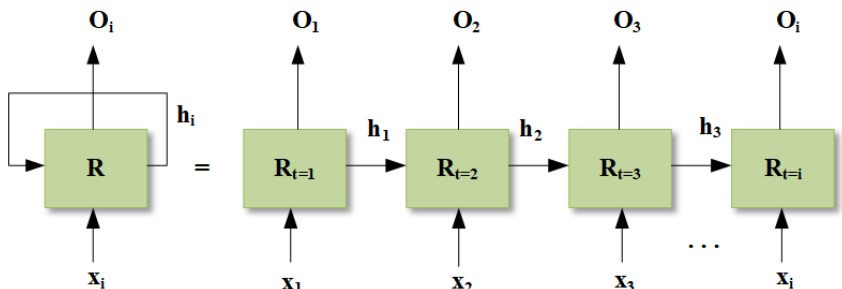

Figure 3: the diagram of unfolded recurrent neural network.

## 3.3 Environment Predictor

As for the Transition function $s_t \sim p(\hat{s}_{t+1}|s_t, a_t)$, we model it with a LSTM to predict the future possible states which is a recurrent model as showed in Fig. 3. We combine the current action made by

action model $a_t$, the state $s_t$ encoded by VAE, and the LSTM hidden state to generate the prediction of future state. To provide more information to down stream action model we use an extra fully connection layer to predict the future reward $p(r_t|s_t, a_t)$ given current state and action.

### 3.4 Action and Value Model

Consider imagined trajectories with a finite horizon H. We uses an actor critic approach to learn behaviors that consider rewards in the future. We learn an action model and a value model in the latent space of the world model for this. The action model implements the policy and aims to predict actions that solve the actual environment. The value model estimates the expected rewards that the action model achieves from each state $s_t$. For simplicity and efficiency, we use two individual linear models to model the action $a_t = W[s_t, \hat{s}_{t+1}] + b$ and value $v_t = W[s_t, \hat{s}_{t+1}] + b$, respectively.

## 4 Learning the behaviours

### 4.1 World Model Learning

Following the derivation of Hafner et al. [2019b], we define the Variational Information Bottle-neck(VIB) objective Tishby et al. [2000] for representation learning of the latent dynamics models. Considering that models trained with the VIB objective outperform those that are trained with other forms of regularization, in terms of generalization performance and robustness to adversarial attack. The objective is shown below,

$$\max I(s_{t,t+1}; (o_{t,t+1}, r_{t,t+1})|a_{t,t+1}) - \beta I(s_{t,t+1}, i_{t,t+1}|a_{t,t+1}), \tag{6}$$

$$p(o_t|i_t) \doteq \theta(o_t - \bar{o}_t), \tag{7}$$

where $\beta$ is scalar and $i$ are dataset indices that determine the observations as in Alemi et al. [2016]

Maximizing the objective leads to model states that can predict the sequence of observations and rewards while limiting the amount of information extracted at each time step.

For the generative objective, we lower bound the first term using the non-negativity of the KL divergence and drop the marginal data probability as it does not depend on the representation model,

$$
\begin{aligned}
I(s_{t,t+1}; (o_{t,t+1}&, r_{t,t+1})|a_{t,t+1}) \\
&= E_{p(s_{t,t+1}, o_{t,t+1}, r_{t,t+1}, a_{t,t+1})}(\sum_t \ln p(o_{t,t+1}, r_{t,t+1}|s_{t,t+1}, a_{t,t+1}) \\
&\quad - \ln p(o_{t,t+1}, r_{t,t+1}|a_{t,t+1})) \\
&\overset{\pm}{=} E(\sum_t \ln p(o_{t,t+1}, r_{t,t+1}|s_{t,t+1}, a_{t,t+1})) \\
&\leq E(\sum_t \ln p(o_{t,t+1}, r_{t,t+1}|s_{t,t+1}, a_{t,t+1})) \\
&\quad - KL(p(o_{t,t+1}, r_{t,t+1}|s_{t,t+1}, a_{t,t+1})|| \prod_t q(o_t, s_t)q(r_t|s_t)) \\
&= E(\sum_t \ln q(o_t|s_t) + \ln q(r_t|s_t))
\end{aligned}
\tag{8}
$$

As for the Transition function of our model, Since it is a probabilistic model whose outputs are a mixture Gaussian distribution with 5 components. Therefore, we train this model with negative log likelihood loss as our prediction model objective.

$$\mathcal{L}_{gmm} = -\sum \sum_{i=1}^{5} \ln \alpha_i \mathcal{N}(\mu_i, \sigma_i), \tag{9}$$

where $\alpha_i$ is the weight of each components, $\mu$ and $\sigma$ are the output of LSTM.

## 4.2 Policy Learning

We will now describe the details of policy learning. We use the proximal policy optimization (PPO) algorithm Schulman et al. [2017] with $\gamma = 0.95$. The algorithm generates rollouts in the simulated environment $env'$ and uses them to improve policy $\pi$. The fundamental difficulty lays in imperfections of the model compounding over time. To mitigate this problem we use short rollouts of $env'$. Therefore we only feed $s_t$ and predicted next state $\hat{s}_{t+1}$ to our actor-critic model. There is a alternative possible solution, which we restart $env'$ from a ground truth every N steps. Using short rollouts may have a degrading effect as the PPO algorithm does not have a way to infer effects longer than the rollout length. To ease this problem, in the last step of a rollout we add to the reward the evaluation of the value function. Training with multiple iterations restarting from trajectories gathered in the real environment is new to our knowledge, and we will have some exploration in this point of view.

# 5   Experiments

Our model is evaluated on gym environment, which is a toolkit for developing and comparing reinforcement learning algorithms. It supports teaching agents everything from walking to playing games like Pong or Pinball. We will train our model in cCarRacing-v0 environment and observe if the car move smoothly and safely along the trail. We find this task interesting because although it is not difficult to train an agent to wobble around randomly generated tracks and obtain a mediocre score. However to get a higher score the agent can only afford very few driving mistakes.

## 5.1 Implementation Details

We based on the code framework provide by TA, and follow the Ha and Schmidhuber [2018] to construct the VAE and LSTM to warp the environment. The PPO algorithm between the vanilla PPO is the different of input. Other hyperparameters of policy learning are same with the vanilla PPO. We set learning rate as $1e - 4$, max norm as $100$.

World model is first pre-trained for one epoch on the data generated by random actions. Then Model is jointly train with the PPO in a very slow update frequency. Namely, every ten sample iteration we calculate and backpropagate the loss of the world model. The benefits of this setting is that We can reduce the time consuming in the updating the parameter of world model part and guide world model focus on task-relevant areas of the image.

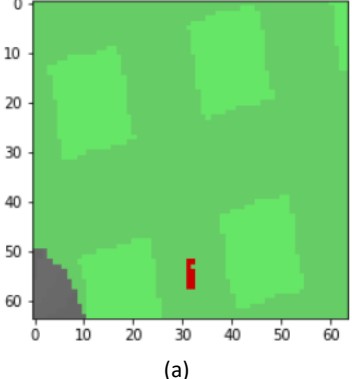
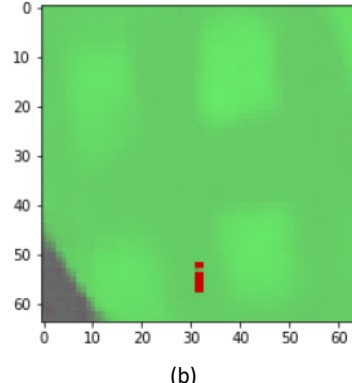

(a)                                                    (b)

Figure 4: The result of reconstruction, (a) is the original observation retrieved from the environment and (b) is the reconstruction results, the key element of the observation have been retained in (b). The blurs in (b) is caused by the average effect of VAE.

## 5.2   Representation Learning

We designed a VAE network to encode the observation from sensors. First is the definition of our layers. 4 convolutions that are mapped onto 2 linear vectors representing the mean and the standard deviation of our VAE. Then another linear layer is added that takes the output (mean, std) and maps it to a vector that will be the input of the decoder. We feed the observation form the environment into our trained VAE and obtain its latent representation as in Fig 2. We can found that our VAE capture the task-relevant information such as the tile of racing track. To further increase the quality of the reconstruction results the task we can add a reward loss to regularized the learning gradient of the VAE.

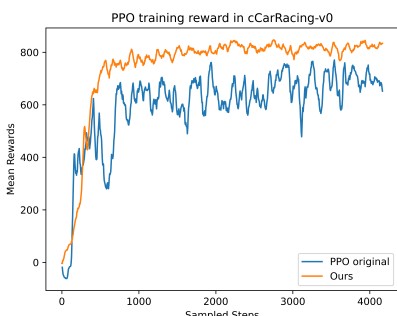
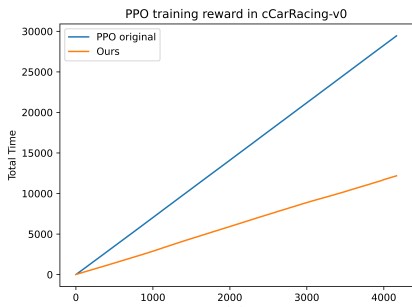

Figure 5: the performance of PPO and our method in the respect of mean reward. Compare to Vanilla PPO our method is more stable and reach the best mean reward in a very short time. this indicate the data efficiency of our method

Figure 6: The total time variation during the training, our method only need one third of time to sample and update model compare with the vanilla PPO in each step.

## 5.3   data efficiency

The primary evaluation in our experiments studies the sample efficiency of our method, in comparison with vanilla PPO. The results of the comparison are presented in Fig 5.

Our method outperforms the model-free algorithms in terms of learning speed on the environment we provided. In particular, it reaches the same performance that our PPO implementation reaches at 10M steps. This indicates that our world model-based methods provides an effective approach to help PPO to learning the Car Racing problem, at a fraction of the sample complexity.

The mean reward of our method along time horizon in the Fig 5 is more stabilized than vanilla PPO. Stabilizing performance means the policy model can easily find a robust optimal point by training inside our world model simulated environment. But other side of the coin is that the capacity of the world model will constrain the model capability, since the domain mismatch between the model and the real environment. This suggests that further improve the quality of state representation of world model would improve the final performance, indicating an important direction for future work.

In addition, the total training time of the our method is one third of vanilla PPO as shown in the Fig **??**. This is benefited by our simple policy model which only have 4160 parameters. Besides, our simple and efficient policy model may enable our method the tackle the larger and more sophisticate environment which is impossible to use to vanilla PPO such as the real world autonomous driving.

# 6 Conclusion

In this paper we proposal a simple and efficient world-model method improve the performance of PPO. For this, we propose an actor critic method that learning a parametric policy through the learned latent dynamics. Our method outperforms vanilla PPO methods in data-efficiency, computation time, and final performance on a variety of challenging continuous control tasks with image inputs. Future research on representation learning can likely scale latent imagination to environments of higher visual complexity.

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
