# OpenReview forum: "Increasing Data Efficiency of Driving Agent By World Model"
_CUHK.edu.hk/2021/Course/IERG5350_

### Official Review · AnonReviewer2 · 2020-12-15
**A simple and efficient application of world model.**

**Rating:** 6
**Confidence:** 4

**Review:**

Summary:
This paper propose to apply the World Model on PPO algorithm to improve the data efficiency of such model-free methods so that it accelerates the RL training process. Specifically, they use VAE to compress the observed image input and use LSTM model to learn the transition function. Experimental results on the simple environment cCarracing-V0 shows the ability of the proposed method to improve training efficiency.

Comments:
1. It is a simple but efficient application of world model on the PPO algorithm to improve its efficiency. But I think it may be a bit insufficient for their contributions since it is just a simple application of the existing methods.
2. The conclusion of the paper's key contributions seems unclear for there are some overlaps among three points. For me, the first point has covered most parts of the other two points.
3. There are some grammar mistakes so that I fail to understand some of the sentences, e.g. "The future discounted rewards is optimize Bellman consistency for latent rewards."
4. There is a typo in the last paragraph of section 5, which shows as "??".

---

### Official Review · AnonReviewer1 · 2020-12-18
**Good Idea to increase sample efficiency for RL**

**Rating:** 7
**Confidence:** 4

**Review:**

Summary: In this paper, the author proposed a novel method to combine model free RL algorithm(PPO) with model based prediction method. Firstly, they compress the original observation into vector via VAE. To increase sample efficiency, they then predict the future observation with learned world model and integrate this prediction into policy and value function.

Pros:
1. Good idea to use VAE to compress the image, making it simple for future prediction and fasten computation
2. Quite interesting idea to model the policy and value function based on current state and future predicted state

Cons:
1. In Section 4.1, it’s quite hard to get the idea of the method. What do you mean by saying i are the dataset indices that determine the observations? Do you index all the observation? Besides, what’s the meaning of q function q(ot,st), it’s a learned probability distribution or what? It’s better to clarify all these unexplained terms.
2. In Section 4.1, when training environment predictor, why the mixture Gaussian distribution has 5 components? What’s the specific meaning of these 5 components? The author hasn’t clarified it.
3. Some typo. Like the fig ?? In page 8.
4. For fig 5 and fig 6, it’s quite hard to read the words below since they are mixed together. And the title of fig 6 is the same with fig 5, while fig 6 should have shown the computation time spent.
5. The reference format is informal. No index.

Suggestion and Question:
1. Is it good enough to only predict the state in next frame? If the speed of the car is quite high, maybe we should predict the state in a longer term to avoid driving mistakes in time?
2. Do you train the VAE and PPO network separately? And when you train the PPO, do you fix the VAE network? Would a method with end-to-end training perform better?

---

### Official Review · AnonReviewer3 · 2020-12-20
**A simple yet effective extension of the model-free solution to cCarRacing environment**

**Rating:** 6
**Confidence:** 4

**Review:**

Summary: The paper tackles the inefficiency of the baseline PPO solution to the CarRacing environment due to its model-free structure, and thus proposes a world model with compressed spatio-temporal representations. It models the environment dynamics from experience by VAE and allows to learn behaviors from imagined outcomes to increase sample-efficiency. Experimental results show that their method effectively increases the training stability and reduces the training time.

Comments:
1. Overall, the paper is well-structured and easy to follow. The evaluation metrics cover a reward curve, time cost, and visualization results of the reconstructed outcome. It's a good extension of the baseline PPO solution and proved to be effective.
2. The figures may not be clearly presented enough. For example, Fig. 1 lacks explanations of the elements it contains, Fig. 2 is a little bit coarse, and Fig. 3 seems to be a general RNN model where the notations are not well aligned with other parts of the paper.
3. There're some typos, missing punctuations (e.g. above Fig. 3 and above Sec. 5.3).
4. I wonder if a tournament could be held between the vanilla PPO trained agent and your agent to further evaluate its performance?